# Merging multi-omics with proteome integral solubility alteration unveils antibiotic mode of action

**Ritwik Maity[1,2,3], Xuepei Zhang[4,5,6], Francesca Romana Liberati[7], Chiara Scribani Rossi[7], Francesca Cutruzzolá[7], Serena Rinaldo[7], Massimiliano Gaetani[4,5,6], José Antonio Aínsa[1,3,8,9], Javier Sancho[1,2,3]\***

[1]Biocomputation and Complex Systems Physics Institute (BIFI)-Joint Unit GBsC-CSIC, University of Zaragoza, Zaragoza, Spain; [2]Departamento de Bioquímica y Biología Molecular y Celular, Faculty of Science, University of Zaragoza, Zaragoza, Spain; [3]Aragon Health Research Institute (IIS Aragón), Zaragoza, Spain; [4]Department of Medical Biochemistry and Biophysics, Karolinska Institutet, Stockholm, Sweden; [5]Chemical Proteomics Unit, Science for Life Laboratory (SciLifeLab), Stockholm, Sweden; [6]Chemical Proteomics, Swedish National Infrastructure for Biological Mass Spectrometry (BioMS), Stockholm, Sweden; [7]Department of Biochemical Sciences "A. Rossi Fanelli", Sapienza University of Rome, Rome, Italy; [8]Departamento de Microbiología, Pediatría, Radiología y Salud Pública, Faculty of Medicine, University of Zaragoza, Zaragoza, Spain; [9]CIBER de Enfermedades Respiratorias—CIBERES, Instituto de Salud Carlos III, Madrid, Spain

*For correspondence: jsancho@unizar.es

Competing interest: The authors declare that no competing interests exist.

**Abstract** Antimicrobial resistance is responsible for an alarming number of deaths, estimated at 5 million per year. To combat priority pathogens, like *Helicobacter pylori*, the development of novel therapies is of utmost importance. Understanding the molecular alterations induced by medications is critical for the design of multi-targeting treatments capable of eradicating the infection and mitigating its pathogenicity. However, the application of bulk omics approaches for unraveling drug molecular mechanisms of action is limited by their inability to discriminate between target-specific modifications and off-target effects. This study introduces a multi-omics method to overcome the existing limitation. For the first time, the Proteome Integral Solubility Alteration (PISA) assay is utilized in bacteria in the PISA-Express format to link proteome solubility with different and potentially immediate responses to drug treatment, enabling us the resolution to understand target-specific modifications and off-target effects. This study introduces a comprehensive method for understanding drug mechanisms and optimizing the development of multi-targeting antimicrobial therapies.

## eLife assessment

This **fundamental** study provides insights into how pathogens respond, on a systemic level including several gene targets and clusters, to selected antimicrobial molecules. **Compelling** evidence is provided, through multi-omics and functional approaches, that very similar molecules originally designed to target the same bacterial protein act differently within the context of the whole set of cellular transcripts, expressed proteins, and pre-lethal metabolic changes. Given the rapid accumulation of omics data and the much slower capacity of extracting biologically relevant insights from big data, this work exemplifies how the development of sensitive data analysis is still a major necessity in modern research.

## Introduction

Antimicrobial resistance poses a global threat of profound proportions. It is estimated that drug-resistant infections contribute to nearly 5 million deaths every year (*Murray et al., 2022*). The development of new, ideally narrow-spectrum antibiotics for 'Priority pathogens' is one of the key factors in reducing the toll, along with international cooperation, vaccines, and diagnostics. *Helicobacter pylori* is one of the high-priority pathogens (*World Health Organization, 2017*) in the World Health Organization (WHO) list of bacteria for which new antibiotics are urgently needed. A meta-study published in 2018 has estimated *H. pylori* infection has a prevalence of 44.3% worldwide (*Zamani et al., 2018*). The eradication of *H. pylori* is carried out using either triple or quadruple chemotherapy, in which several antibiotics and antimicrobial compounds including bismuth are used in combination with a proton pump inhibitor. A recent meta-analysis from published US studies estimated the individual rate of resistance for clarithromycin, metronidazole (MNZ), and levofloxacin to be higher than 30% (*Ho et al., 2022*). This urges the development of new, ideally narrow-spectrum, antibiotics for the treatment of drug-resistant *H. pylori*. Essential metabolic pathways are a vast promising scenario for antimicrobial development. Flavodoxin, urease, menaquinone synthesis enzymes, and respiratory complexes are among the most promising potential disease-specific targets for obtaining narrow-spectrum anti-*H. pylori* agents (*Salillas et al., 2021*; *Cunha et al., 2021*; *Lettl et al., 2023*). In this study, we have used five compounds developed from a high-throughput screening conducted to identify inhibitors against *H. pylori* flavodoxin (Hp-Fld) (*Salillas et al., 2019*). Hp-Fld is an essential protein for *H. pylori*, and it is not present in humans as an independent protein, which makes it a preferred candidate for anti-*H. pylori* drug development. The selected compounds (*Figure 1—figure supplement 1A*) derive from 4-nitrobenzoxadiazole (IV and IVj) or 4-amine benzoxadiazole (IVa and IVk) and differ in terms of solubility and antimicrobial specificity spectrum. IVb is a hydrophobic analog not having a benzoxadiazole moiety.

The identification of the modes of action of new potential drugs is a major challenge in the chemical systems biology of diseases and there are numerous bioactive compounds (including approved drugs) whose mechanisms of action remain unknown (*Iwata et al., 2017*; *Santos et al., 2017*). Whether identifying the mode of action of a new antibiotic is a medical priority can be debated. However, a mechanistic understanding at the molecular level is of immense benefit to its future development as a drug. The mode of action of an antibiotic is generally classified depending on the interaction with its target, which is often a protein essential for maintaining bacterial viability. However, as with any drug, an antibiotic may interact not only with its primary target but also with other unexpected targets in the bacterium or the host (off-targets) (*Heppler et al., 2022*). This may cause unforeseen side effects that, although typically undesired, may occasionally enable new therapeutic indications.

Target-based drug discovery pipelines rely on in vitro screening protocols mostly monitoring drug–target (usually a protein) binding or concomitant changes in protein stability. Despite their scalability and adaptivity, these protocols may or may not accurately represent the drug's target and off-target interactions in vivo. Recent developments in omics techniques can provide a molecular-level view of drug-induced changes in cellular processes. The robustness of omics techniques in pathway prediction has been demonstrated and a wide range of life science fields increasingly rely on them. Compared to conventional approaches, omics techniques provide a more holistic molecular perspective on biological systems. While large-scale omics data are becoming more accessible, and multi-omics studies are more frequent, multi-omics integration of data remains challenging. Here, we propose a post-analysis data integration method for pathway prediction and validation of the mechanism of action of Hp-Fld inhibitors. It is a long-standing knowledge that, when heated, proteins denature and generally become insoluble, and interaction with small molecules can induce changes in the thermal stability and solubility of proteins. Thermal shift assay (TSA) (*Molina et al., 2013*; *Mateus et al., 2018*; *Peng et al., 2016*) utilizes this knowledge to understand drug–protein interaction by observing the target's melting temperature shift upon drug treatment. However, when the target is uncertain or the mechanism is to be elucidated, the use of omics techniques must consider different angles to be able to deconvolute changes in the system as a whole upon drug treatment. The deconvolution of unknown targets using TSA-based proteomics approaches (*Molina et al., 2013*) shows some technical biases and a limited throughput to be able to compare multiple drugs in a significant number of biological replicates and to be able to integrate other omics approaches. To address those issues, we adopt here an approach based on the unbiased and high-throughput target deconvolution method known as the

Proteome Integral Solubility Alteration (PISA) assay in the PISA-Express format (*Gaetani and Zubarev, 2023*; *Gaetani et al., 2019*; *Zhang et al., 2022*; *Sabatier et al., 2021*). It is based on the fact that protein targets, as well as other early mechanistic proteins, may undergo changes in solubility as a result of drug binding or changes in related molecular interactions and macro complexes within a cell.

The information obtained on the protein targets of the individual compounds has been utilized to filter the differential expression data from proteomics and transcriptomics, aiming to identify the most influenced pathways. Subsequently, we validated selected pathways at a functional level.

## Results

### Changes in transcriptomics and proteomics landscape, and associated gene ontology

Our investigation into understanding the mode of action of nitro-benzoxadiazole compounds commenced with a comparison of the conventional transcriptional and translational changes induced by these compounds, the vehicle control (DMSO) dimethyl sulfoxide, and the commercially used drug MNZ. RNA sequencing (RNA-seq) and expressional proteomics were employed to identify transcriptional and translational changes, respectively. A total of 771 differentially expressed transcripts (*Figure 1—figure supplement 1D*) and 113 (*Figure 1—figure supplement 1E*) differentially expressed proteins were selected using a linear model fitting analysis followed by a cutoff-based selection. Overall, we have achieved more than 90% of genomic coverage and more than 25% of the transcripts and 10% of the proteins were found to be differentially expressed upon treatment except for Compound IVk, which exhibited 372 differentially expressed proteins in spite of the fact that only 11 transcripts showed differential expression at a p-value of 0.001 and 175 at a p-value of 0.05. Lack of correlation between mRNA and protein expressions has been reported before (*Liu et al., 2016*; *Bathke et al., 2019*; *Haider and Pal, 2013*). It has been related to differences in half-lives, post-transcription machinery, and transcriptional and translational decoupling in bacteria. The differentially expressed *H. pylori* transcripts in the presence of the individual drugs were investigated using gene set enrichment analysis (*Figure 1—figure supplement 2A–G*).

We compared transcriptomics and proteomics data for the lead Compound IV (*Figure 1A*) and found 174 common IDs (gene or protein). These IDs show enrichment in gene ontology (GO) terms 'Protein containing complex', 'Cellular anatomical entity', 'Proton-transporting ATP synthase complex', and 'Non-membrane-bounded organelle'. Two GO terms, 'Metabolic process (GO:0008152)' and 'Organonitrogen compound biosynthetic process (GO:1901566)', are present in all compounds except for Compound IVk (*Figure 1E*). Some analogies for pairs of compounds sharing either the nitro, the amine group, or the polar moiety stand out. For example, the 'Proton-transporting ATP synthase complex (GO:0045259)' is found in the nitro-containing Compound IV (*Figure 1A*), while its soluble derivative, Compound IVj, is the only compound containing the 'ATP metabolic process (GO:0046034)' and 'ATP synthesis coupled proton transport (GO:0015986)' (*Figure 1D*), suggesting IVj exerts a specific effect on ATP production and utilization. ATP-synthase-related functions are thus relevant in the two closely related nitro compounds tested. On the other hand, both Compound IVa (*Figure 1B*) and its soluble derivative IVk (*Figure 1E*) share the ontology term related to gene expression (GO:0010467), highlighting their potential alteration of this biological process. The GO term 'cellular process (GO:0009987)' is present in the sets of the two relatively more soluble compounds: IVj and IVk. This ontology covers a broad range of cellular processes, including cell-to-cell communication. The gene ontologies (*Figure 2D*) associated to Compound IVb – the hydrophobic compound lacking the nitro/amine-benzoxadiazole moiety – are related to those of IVa, sharing several GO terms related to metabolic processes, nitrogen compound biosynthesis, and organonitrogen compound biosynthesis. Additionally, IVb includes GO terms related to heterocycle biosynthesis.

### PISA and determination of intracellular targets

The effectiveness of an antimicrobial agent is often determined by its ability to specifically interact with a cellular target of the microbe. Very few techniques have demonstrated the ability to identify targets at a system biology level. Here, the PISA assay has been used to identify intracellular targets for Compound IV and its derivatives. This is the first time PISA assay, in the form of PISA-Express, has been successfully performed in living bacterial cells, with protocols adapted and modified from

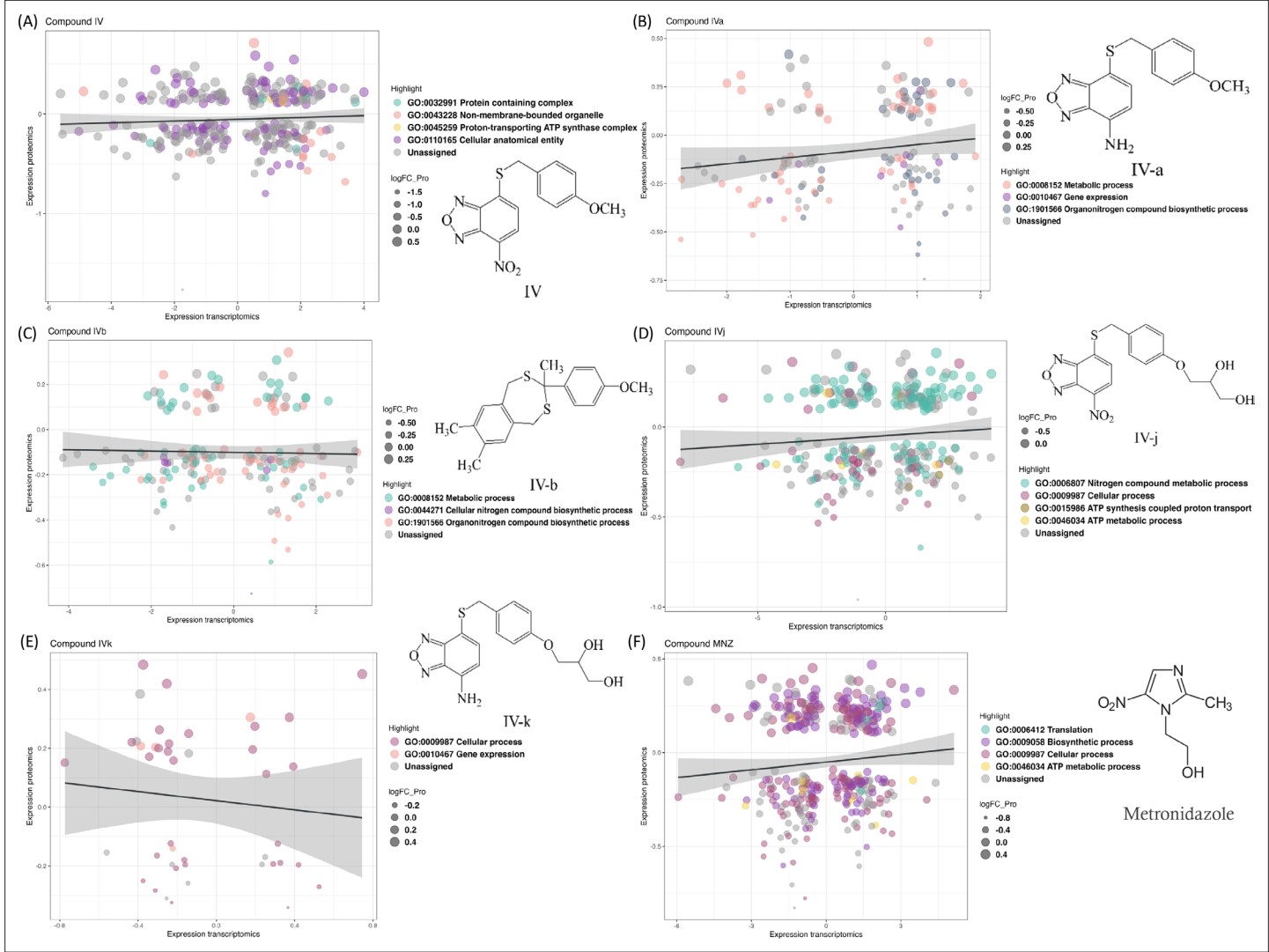

**Figure 1.** Changes in the transcriptomics and proteomics landscape, along with the associated gene ontology. (**A–F**) These panels highlight the common genes and proteins associated with individual compounds, with different colors indicating the distribution of these genes and proteins across various gene ontology terms.

The online version of this article includes the following figure supplement(s) for figure 1:

**Figure supplement 1.** Compounds used in the assay and the study design.

**Figure supplement 2.** Gene ontology analysis of Compound IV and its derivatives.

previous PISA studies in mammalian cells (**Sabatier et al., 2021**). The experimental procedure involved exposing the bacteria to two different drug concentrations for 20 min (**Figure 2** and **Figure 2—figure supplement 1**). Afterwards, the cells in each sample were treated and split in two parts: one part was subjected to a temperature gradient and the other one was left without heating. Then, the soluble proteins were identified and quantified using mass spectrometry (**Figure 1—figure supplement 1D**). Because of the fast-doubling time of bacteria and related proteome expression changes even in a short time of incubation with compounds, the PISA assay was used in the PISA-Express form where PISA results on protein soluble amounts are normalized pairwise on the total protein abundance changes of the same samples relative to controls of PISA and expression assay.

We have observed several notable trends in the alteration of protein solubility. The top 10 proteins with the most altered stability for each compound are listed in **Table 1**. In this section, we will focus on these top 10 proteins. Three proteins stand out, CagA, HP_0542, and HP_0543, that are actively

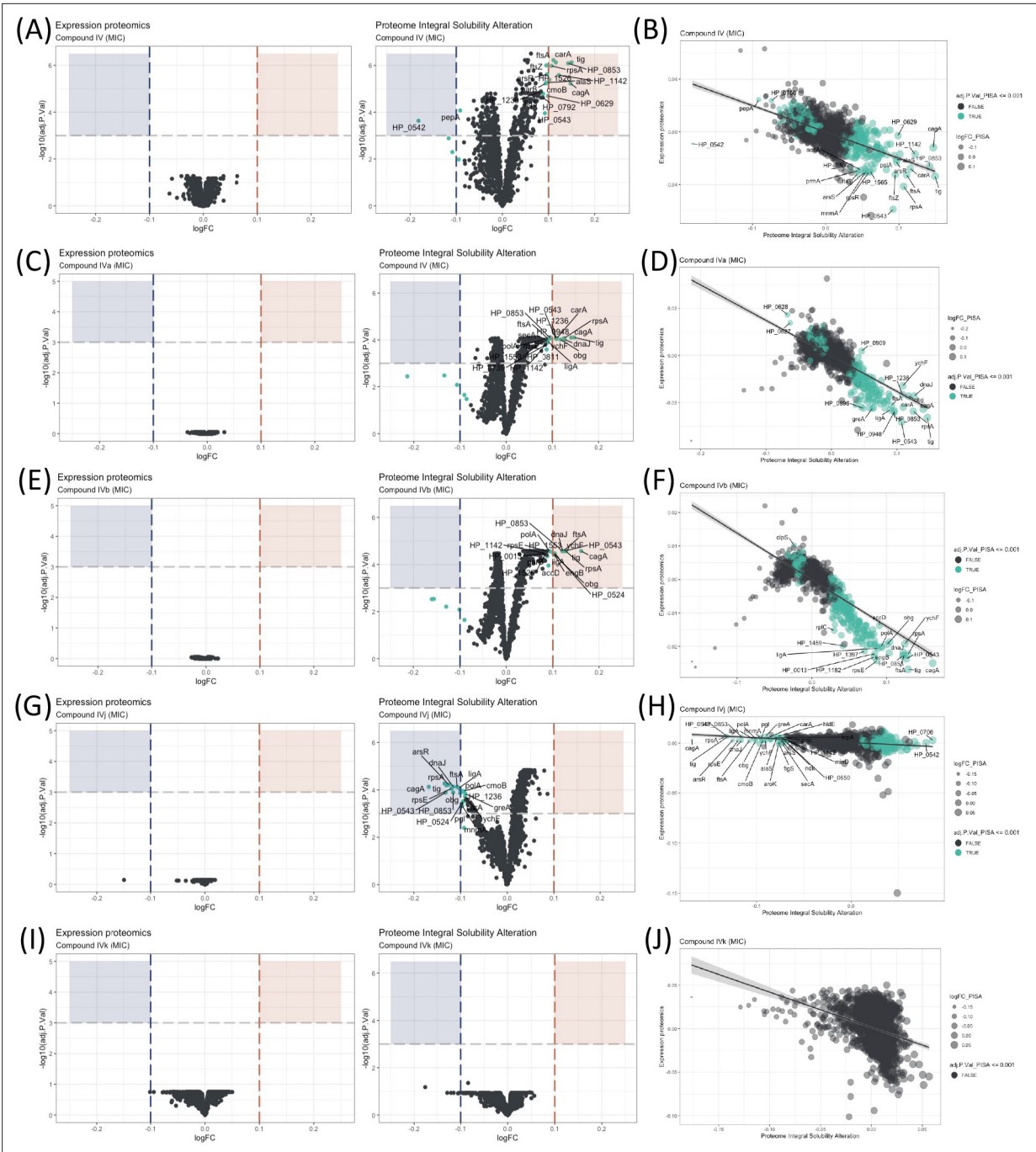

**Figure 2.** Proteome Integral Solubility Alteration (PISA) analysis where significant intracellular target alterations are depicted in teal. For Compounds IV, IVa, and IVb, significant solubility changes indicate protein stabilization, whereas for IVj, the significant solubility changes indicate destabilization. (**A**) Volcano plot of protein solubility alterations for Compound IV at minimum inhibitory concentration (MIC). (**B**) Scatterplot showing protein solubility changes for Compound IV compared to baseline. (**C**) Volcano plot of protein solubility alterations for Compound IVa at MIC. (**D**) Scatterplot displaying protein solubility changes for Compound IVa compared to baseline. (**E**) Volcano plot of protein solubility alterations for Compound IVb at MIC. (**F**) Scatterplot indicating protein solubility changes for Compound IVb compared to baseline. (**G**) Volcano plot of protein solubility alterations for Compound IVj at MIC. (**H**) Scatterplot highlighting protein solubility changes for Compound IVj compared to baseline. (**I**) Volcano plot showing no significant alterations in protein solubility for Compound IVk at MIC. (**J**) Scatterplot displaying no significant alterations in protein solubility for Compound IVk compared to baseline.

The online version of this article includes the following figure supplement(s) for figure 2:

*Figure 2 continued on next page*

*Figure 2 continued*

**Figure supplement 1.** Proteome Integral Solubility Alteration coupled with expression proteomics (PISA-Express) analysis for different compounds at 5× of minimum inhibitory concentration (MIC).

**Figure supplement 2.** Concentration-dependent changes in Proteome Integral Solubility Alteration.

involved in the *cag* pathogenicity island renowned for its role in *H. pylori* virulence and for its association with an increased risk of gastric cancer (*Parsonnet et al., 1997*).

Our observations indicate that all our compounds, except IVk, significantly affect the solubility of at least two out of the three above-mentioned proteins. Compounds IV (*Figure 2A, B*) and IVa (*Figure 2C, D*) specifically change the solubility of CagA and HP_0542, while Compound IVj (*Figure 2G, H*) changes the solubility HP_0543 in addition to CagA. Only Compound IVb (*Figure 2E, F*) changes the solubility of all three of these proteins. It is worth mentioning that, like other thermal proteome profiling-based assays, PISA assay can identify changes in the solubility of specific target proteins and in that of the protein partners of their complex, as well as any other changes of solubility due to the altered complex formation and post-translational modifications. Changes in the solubility of multiple proteins within a specific complex strengthen the correlation between the drug and the complex itself. On the other hand, our PISA analysis has identified two proteins, FtsA and FtsZ, associated with the Z-ring formation in cell division that are affected by the compounds. Inhibition of Z ring formation has been a proven target for antimicrobial development. The orally bioavailable methylbenzamide antibiotic TXA-709 and its active metabolite TXA-707 target FtsZ and have been tested (Phase I) against *Staphylococcus aureus* (*Lepak et al., 2015*). Compounds IV and IVb show a clear concentration-dependent targeting of both proteins (*Figure 2—figure supplement 2A, C*), while FtsZ and FtsA are additionally targeted by Compounds IVa and IVj, respectively, albeit without a clear concentration dependency. Another target shared among several compounds is the chaperone protein trigger factor (Tig), which plays a crucial role in facilitating proper protein folding and is indispensable for the survival of bacterial cells. The solubility of this protein has been altered by all the compounds except IVk (*Figure 2I, J*) in a concentration-dependent manner (*Figure 2—figure supplement 2B, D, E*). The possibility of Tig interacting with other proteins destabilized by the drug, along with the influence of the heat gradient during the PISA assay, may introduce potential noise in the data. Further investigation is required to confirm the interaction of the drug with Tig.

We have also identified a few compound-specific proteins having altered stability. Compound IV strongly altered DNA-PolA and HP_1142, which function as DNA helicases, contributing to DNA replication and repair. Compounds IV and IVa altered the stability of PanD and CarA, which are both involved in the pantothenate biosynthesis pathway. Compound IVj targeted ArsR, HP_0543, HP_0853, LigA, and RpsE, involved in interesting pathways. LigA is essential for DNA replication and repair while ArsR is a member of the two-component system ArsS/ArsR that regulates genes involved in biofilm formation and acid adaptation. Compound IVk is the only compound for which we could not detect any statistically significant altered protein with an adjusted p-value (adj.P.Val) of 0.001. However, Compound IVk altered stability of several proteins with a relatively low adj.P.Val (0.005), as depicted in *Figure 2—figure supplement 1I, J*. Notably, it does not share any of the previously described prominent potential targets.

**Table 1.** Proteins with highest solubility alteration from the Proteome Integral Solubility Alteration (PISA) assay.

| Compound | Proteins with highest solubility alteration | | | | | | | | | |
|---|---|---|---|---|---|---|---|---|---|---|
| IV | cagA | carA | ftsA | HP_0542 | HP_0706 | HP_0853 | HP_1142 | panD | rpsA | ig |
| IVa | cagA | carA | dnaJ | HP_0486 | HP_0542 | HP_1236 | bg | rpsA | tig | ychF |
| IVb | cagA | dnaJ | ftsA | HP_0486 | HP_0542 | HP_0543 | HP_0706 | rpsA | tig | ychF |
| IVj | arsR | cagA | dnaJ | ftsA | HP_0543 | HP_0853 | ligA | rpsA | rpsE | tig |
| IVk | amiF | HP_0750 | HP_0803 | HP_0922 | HP_0993 | HP_1091 | HP_1163 | HP_1276 | HP_1390 | rnhA |

## Drugs interactions with essential proteins

Essential proteins have been primary targets for drug discovery. Reportedly, *H. pylori* possess 323 essential genes (*Salama et al., 2004*). More than half of them is cytoplasmic and related to ribosomal proteins, while over 15% encode cell membrane proteins. The function of over 100 of these proteins remains unknown. We have looked for the interaction of our compounds with all 323 *H. pylori* essential proteins and we have detected 20 essential targets common for Compounds IV, IVa, IVb, and IVj (*Figure 3A–H*), and not essential targets for IVk. It should be indicated that most of the unique proteins mentioned in this section are found in the PISA analysis but they are not in the top 10 targets shown in *Table 1*. The 20 common essential targets are mostly associated with cell division (e.g., FtsZ), small subunit ribosomal proteins (RspC, RspE, RspL, RplE, and InfC). Furthermore, we identified a few unique changes for Compound IV (DnaN, involved in DNA tethering and processivity of DNA polymerases, and C694_06445, which could be a functional equivalent of delta subunit of DNA polymerase III). We also identified other essential proteins upon treatment with different compounds: for Compound IVa (YlxH, involved in the placement and assembly of flagella); for Compound IVb (AroK, shikimate kinase and AroC, shikimate synthase); and for Compound IVj (AtpC, AtpF, and AtpG, associated to F-ATPase, VepD, a very prominent target of virulence-associated protein d). Most interestingly, through our PISA analysis we have detected a clear interaction of Compound IVj with flavodoxin (FldA), the protein that was used as initial in vitro target in the drug discovery process of Compound IV, demonstrating the robustness of our experimental strategy.

## Weighted Correlation Network Analysis and selection of drug-associated pathway

Weighted Correlation Network Analysis (WGCNA) (*Langfelder and Horvath, 2008*) is a useful approach for constructing co-expression networks from high-dimensional omics data. One of the advantages of WGCNA is that the resulting co-expression networks are highly informative and can be used as a reference for downstream analyses of other types of omics data. Here, we have used WGCNA to construct a scale-free signed network from the RNA-seq data. We have used topological overlap measures to merge genes into modules (*Figure 4A*). We detected eight major gene modules and assigned distinct colors to each module for the ease of visualization and description (*Figure 4A*). These modules were correlated with different compounds using Pearson correlation. Through p-value analysis, we identified the module that exhibited the strongest correlation with the compounds of interest (*Figure 4B*). To understand the biological distribution of these modules we have looked for their overlap against the *H. pylori* protein–protein association network from the STRING database (*Szklarczyk et al., 2023*). We divided the protein–protein association network into eight separate clusters using a method called 'k-means'. Each cluster represents a group of proteins that have similar levels of co-expression.

The color module 'red' from WGCNA consists of 88 genes and shows up in different biological clusters (*Figure 4C*). The hierarchical clustering of our compounds in this module reveals that Compounds IVb and MNZ affect genes in this module differently from the control (*Figure 4B* and *Figure 4—figure supplement 1A*). The module 'brown' forms the largest cluster, consisting of 420 genes. For this module, Compounds IV, IVb, IVj, and MNZ show downregulation (*Figure 4B* and *Figure 4—figure supplement 1B*). We have not been able to detect any significant GO term enrichment from color module 'magenta' but genes from this module are upregulated for Compounds IV and IVj (*Figure 4B* and *Figure 4—figure supplement 2A*). Meanwhile, the color modules 'greenyellow' (*Figure 4—figure supplement 2B*) and 'pink' (*Figure 4—figure supplement 3A*). significantly associate with multiple GO terms related to chemotaxis and locomotion. The heatmaps show the upregulation of these genes associated with greenyellow and pink modules upon treatment with Compound IVb and MNZ. Module 'turquoise' includes 342 genes. This module is associated with the GO terms related to gene expression and translation among many other metabolic and biosynthetic processes (*Figure 4—figure supplement 2B*). Genes from this module are downregulated by Compounds IV, IVb, IVj, and MNZ (*Figure 4B* and *Figure 4—figure supplement 3B*). The modules 'black' and 'turquoise' are closely related (*Figure 4A*), but module 'black' is specifically associated with ribosomes and protein-containing complexes (*Figure 4—figure supplement 4A*). Compounds IV and IVj are the only ones that downregulate genes in this module (*Figure 4—figure supplement 4A*). Finally, the module 'purple' is linked to the pathogenesis and interspecies interaction (*Figure 4—figure supplement 4B*).

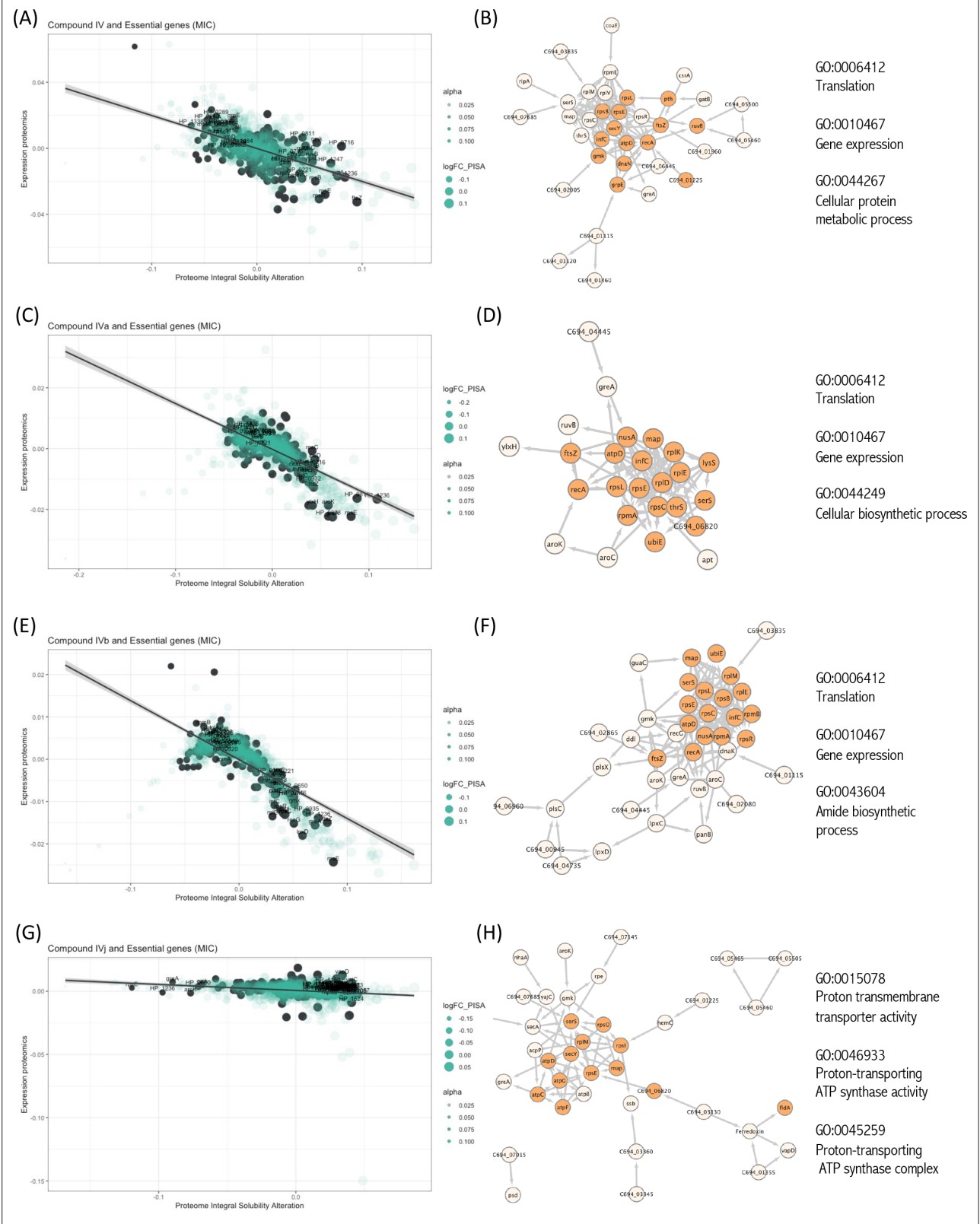

**Figure 3.** Interactions of compounds with essential proteins. Significant alterations in protein solubility are highlighted in black, compared to baseline protein expression changes. The panel does not include Compound IVK as no significant alterations in the solubility of essential proteins were detected for this compound. (**A**) Intracellular essential protein targets detected for Compound IV. (**B**) Protein co-expression network among the essential proteins with significant alterations induced by Compound IV. Highlighted nodes represent the distribution of the primary gene ontology (GO) term associated

*Figure 3 continued on next page*

*Figure 3 continued*

with these proteins. (**C**) Intracellular essential protein targets detected for Compound IVa. (**D**) Protein co-expression network among the essential proteins with significant alterations induced by Compound IVa. Highlighted nodes represent the distribution of the primary GO term associated with these proteins. (**E**) Intracellular essential protein targets detected for Compound IVb. (**F**) Protein co-expression network among the essential proteins with significant alterations induced by Compound IVb. Highlighted nodes represent the distribution of the primary GO term associated with these proteins. (**G**) Intracellular essential protein targets detected for Compound IVj. (**H**) Protein co-expression network among the essential proteins with significant alterations induced by Compound IVj. Highlighted nodes represent the distribution of the primary GO term associated with these proteins.

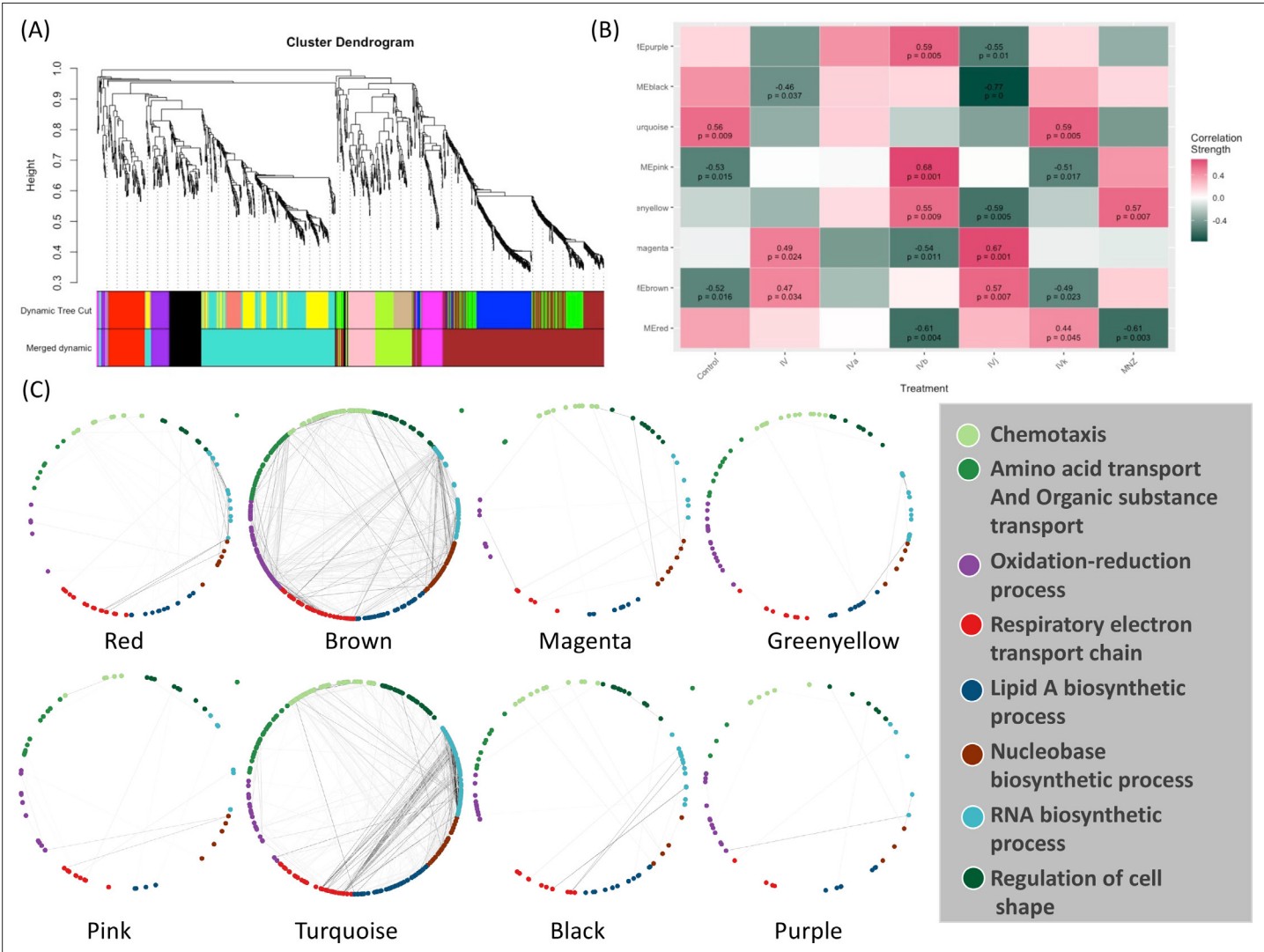

**Figure 4.** Weighted Correlation Network Analysis (WGCNA) and the selection of drug-associated pathways. (**A**) The clustering dendrogram and expression heatmap of genes are presented, identifying the WGCNA modules. (**B**) The correlation between the identified modules and different treatments is displayed. Modules that exhibit a significant association with traits, indicated by a correlation greater than 0.5 and a p-value less than 0.05. Red and green colors represent positive and negative correlations with gene expression, respectively. (**C**) The distribution of different WGCNA modules within the protein co-expression network of *H. pylori* is depicted.

The online version of this article includes the following figure supplement(s) for figure 4:

**Figure supplement 1.** Gene expression heatmap and correlation of different compounds across modules from the WGCNA analysis.

**Figure supplement 2.** Gene expression heatmap and correlation of different compounds across modules from the WGCNA analysis.

**Figure supplement 3.** Gene expression heatmap and correlation of different compounds across modules from the WGCNA analysis.

**Figure supplement 4.** Gene expression heatmap and correlation of different compounds across modules from the WGCNA analysis.

Compounds IV, IVk, and MNZ downregulate the genes associated with this pathway (*Figure 4B* and *Figure 4—figure supplement 4B*).

## Identification of target-associated pathway

We have further integrated the modules identified through WGCNA into Cytoscape, a software platform for visualizing and analyzing complex networks. First, we linked the changes identified through the PISA assay with the modules that showed significant correlations in the WGCNA data. Next, to understand the pathway regulation associated with these changes, we merged the differential protein expression data with this network. This approach enables the detection of changes in gene co-expression associated with changes in solubility through the PISA assay within a specific module and aids in the identification of targets with systemic regulation. In addition, with this approach, it is possible to identify modules which are altered upon drug treatment even if they do not contain any prominent target. We hypothesized that the networks which lack any prominent target may be involved in the cellular response to a situation of stress.

We were able to identify two target groups in the module 'brown'. One group is associated with the virulence factor CagA, while the other is related to the cell division proteins FtsA and FtsZ. Module 'brown' is positively correlated with Compounds IV and IVj and negatively correlated with Compounds IVa (not significant) and IVk (*Figure 4B*). In the same module, several drug targets were found to be related to Compound IV (*Figure 5A*) and, upon close inspection, we detected the downregulation of multiple proteins related to CagA (*Figure 5—figure supplement 1A*). Finally, module 'brown' also includes a few essential protein targets like FtsA and ArsR.

The data from proteomics indicate the downregulation of FtsY associated with FtsA (*Figure 5—figure supplement 1B*). In the case of Compound IVa, no modules were found to have a significant p-value. Therefore, we relied on correlation strength and analyzed the modules 'brown' (*Figure 5B*), 'magenta' (*Figure 5—figure supplement 1G*), and 'purple' (*Figure 5—figure supplement 1H*) in relation to this compound. Notably, we observed a very similar regulation of CagA in the case of Compound IVa (*Figure 5—figure supplement 1E*) when compared to Compound IV. However, the downregulation of FtsY was not observed in this instance. Instead, we observed PolA and RecN as targets (*Figure 5—figure supplement 1F*). It is worth mentioning that a single protein can be shared by closely related modules; in this instance, FtsA is present in both the modules 'brown' and 'pink'. Compound IVb shows a significant correlation with the module 'pink'. We have detected FtsA as a potential target for Compound IVb and in this analysis multiple proteins related to RNA processing, and associated with FtsA were found downregulated in this module (*Figure 5C*). Compound IVj shows five correlated modules and three of them are very similarly regulated modules compared to Compound IV. In module 'brown', we have detected six potential targets which include FtsA, PolA, and CagA (*Figure 5E*). Finally, while we have not detected any targets in module 'pink' or 'brown' associated with Compound IVk, FtsA is upregulated in both modules (*Figure 5—figure supplement 2F, G*).

Compound IV does not have any target in the module 'magenta', but RimP, which is involved in 30S ribosomal subunit maturation, is prominently downregulated in this module (*Figure 5—figure supplement 1C*). The module 'black' associated with this compound contains Tig, which is involved in facilitating proper protein folding, as a target. Tig downregulates multiple proteins associated closely with S12 ribosomal protein of the 30S subunit (*Figure 5—figure supplement 1D*) indicating its involvement in stabilization of ribosomal protein. In the Compound IVa-associated module 'magenta', we identified ObgE, which is an essential GTPase, as a target (*Figure 5—figure supplement 1G*). However, no specific target was detected for the module 'purple', except for FliG, which is the flagellar motor switch protein. This protein was found to be downregulated with Compound IVa treatment (*Figure 5—figure supplement 1H*). Protein ObgE is also one of the prominent candidate targets associated with Compounds IVb and IVj in the module 'magenta' and we have detected the downregulation of multiple proteins related to ABC transporter (*Figure 5—figure supplement 2B, D*). We have not identified any significant targets in the modules 'purple', 'greenyellow', and 'red' that associate with Compound IVb (*Figure 5—figure supplement 1I, J*, *Figure 5—figure supplement 2A*) and modules 'purple', 'black', and 'greenyellow' which associate with IVj (*Figure 5—figure supplement 2C–E*). In the case of Compound IVk, we have detected four highly correlated modules, including the module 'turquoise', which is unique to this compound. Within these modules, we have

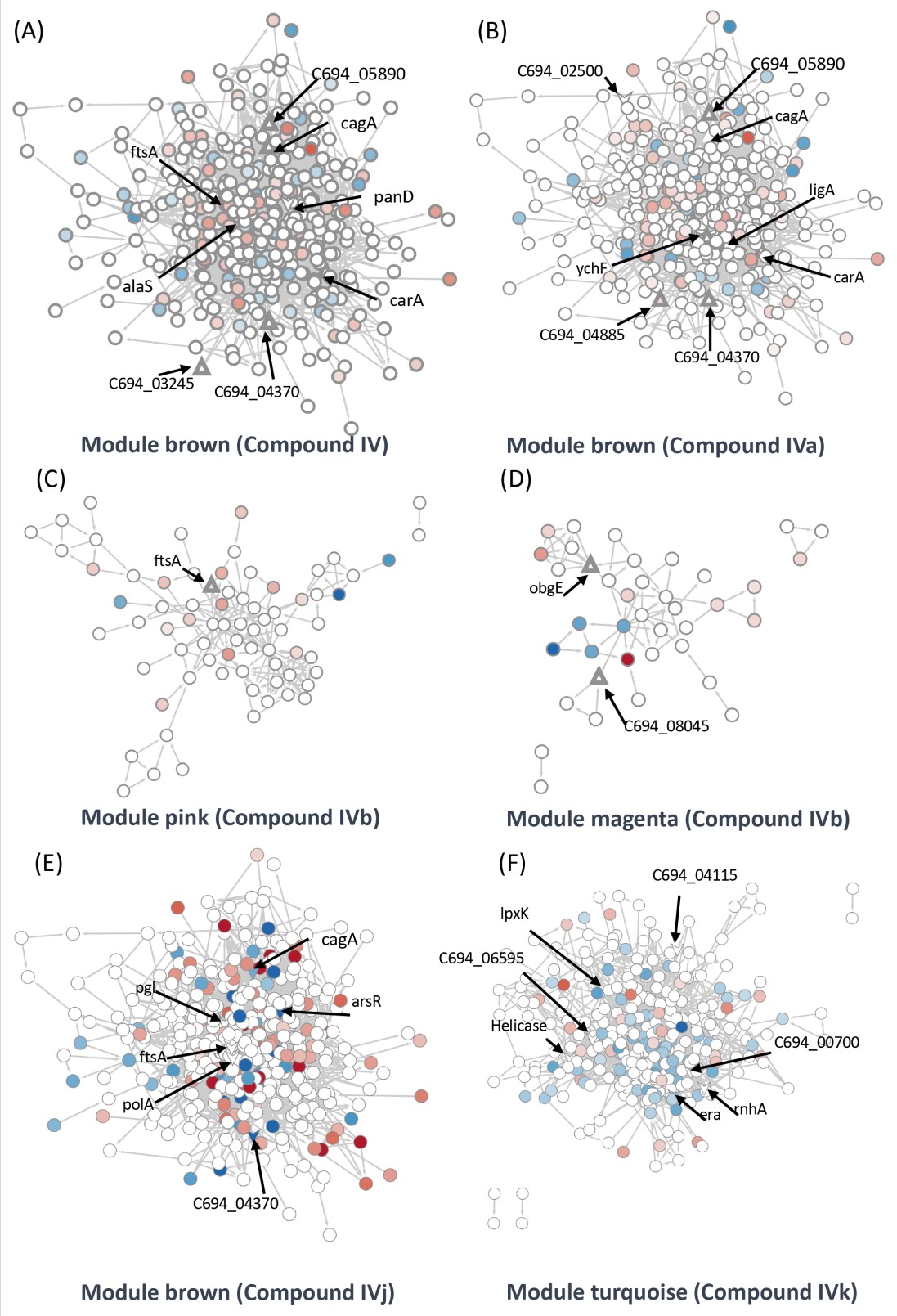

**Figure 5.** Identification of target-associated pathways. (**A**) Differential expression of proteins influenced by Compound IV. Upregulated proteins are represented in red, while downregulated proteins are represented in blue. The targets associated with Compound IV are highlighted within the brown module (triangle). (**B**) Differential expression of proteins influenced by Compound IVa. Upregulated proteins are represented in red, while downregulated proteins are represented in blue. The targets associated with Compound IVa are highlighted within the brown module (triangle).

*Figure 5 continued on next page*

(**C**) Differential expression of proteins influenced by Compound IVb. Upregulated proteins are represented in red, while downregulated proteins are represented in blue. The targets associated with Compound IVb are highlighted within the pink module (triangle). (**D**) Differential expression of proteins influenced by Compound IVb. Upregulated proteins are represented in red, while downregulated proteins are represented in blue. The targets associated with Compound IVb are highlighted within the megenta module (triangle). (**E**) Differential expression of proteins influenced by Compound IVj. Upregulated proteins are represented in red, while downregulated proteins are represented in blue. The targets associated with Compound IVj are highlighted within the brown module (triangle arrow symbol). (**F**) Differential expression of proteins influenced by Compound IVk. Upregulated proteins are represented in red, while downregulated proteins are represented in blue. The targets associated with Compound IVk are highlighted within the turquoise module (triangle arrow symbol).

The online version of this article includes the following figure supplement(s) for figure 5:

**Figure supplement 1.** Associations between different modules and specific targets for Compound IV and its derivatives.

**Figure supplement 2.** Associations between different modules and specific targets for Compound IV and its derivatives.

identified seven potential targets, including helicase, RnhA, and GTPase Era (*Figure 5F*). Module 'turquoise' is highly associated with gene expression, and the imbalance in differential transcripts and proteins may indicate transcriptional and translational decoupling, warranting further study for confirmation. Additionally, in modules 'brown' and 'red', we have identified two potential targets as hypothetical proteins associated with Compound IVk (*Figure 5—figure supplement 2F–H*). In several of our studies, Compound IVk, which has a higher MIC, exhibits markedly different behavior. This difference in behavior may stem from different sources, including intercellular availability, inactivation inside the cell, or loss of target specificity. Multiple studies have previously demonstrated that there is only a 30% chance for a structurally similar compound to have similar biological activity (*Martin et al., 2002*).

## Biophysical assays to validate changes in targeted pathway

Elucidation of the mode of action of antimicrobial drugs has evolved in recent years. There is still considerable debate related to the role of reactive oxygen species (ROS) and their link to DNA damage in bactericidal drug-mediated antimicrobial activity. The investigation of the role of ROS as an overarching mode for bactericidal drug-mediated antimicrobial activity has led to some conflicting evidence either showing a common link between ROS and bactericidal drug-mediated antimicrobial activity (*Kohanski et al., 2007*), or demonstrating that the bactericidal activity does not depend on ROS (*Keren et al., 2013*; *Liu and Imlay, 2013*). As our group of compounds exhibits a bactericidal behavior (*Figure 6A*), we have looked for evidence of ROS generation. Compound IV and its derivatives cause a marked increase in ROS generation when compared to the control (DMSO). Though a degree of heterogenicity is expected due to the cellular stages, distribution and penetration of the drug at an early phase, Compounds IV and IVj exhibit a broader change in ROS production (*Figure 6B*). This indicates that the nitro-bearing groups have a higher propensity to generate ROS. We have also observed that the genes associated with the generation of ROS are significantly overexpressed for Compounds IV, IVb, IVj, and MNZ (*Figure 6—figure supplement 1A*). As described above and depicted in *Figure 6—figure supplement 1B*, multiple DNA damage repair proteins and genes are downregulated in the presence of Compounds IV, IVb, IVj, and MNZ. Additionally, DNA PolA was found to be a major target for Compound IVj. Following these results, we investigated compound-induced DNA damage using the APO BrdU TUNEL assay. All the compounds, particularly IV and IVj, caused significant DNA damage (*Figure 6C*).

On the other hand, given that these drugs indicated involvement of multiple factors from the electron transport chain including flavodoxin and we observed significant drop in the ATP production rate (*Figure 6D*) associated to Compounds IV and IVj, we have investigated the changes in oxygen consumption rate (OCR) as we hypothesize that a reduction in soluble flavodoxin could lead to decreased OCR. Though the signal-to-noise ratio of these data is poor and further evaluation to quantify the amount of reduction in oxygen consumption is needed, the data suggest a reduction in OCR to a significant extent for Compounds IV, IVa, and IVj at corresponding MIC (*Figure 6E*). Finally, we also have been able to detect a significant reduction in OCR at a higher concentration (2× MIC) of MNZ and Compound IVk.

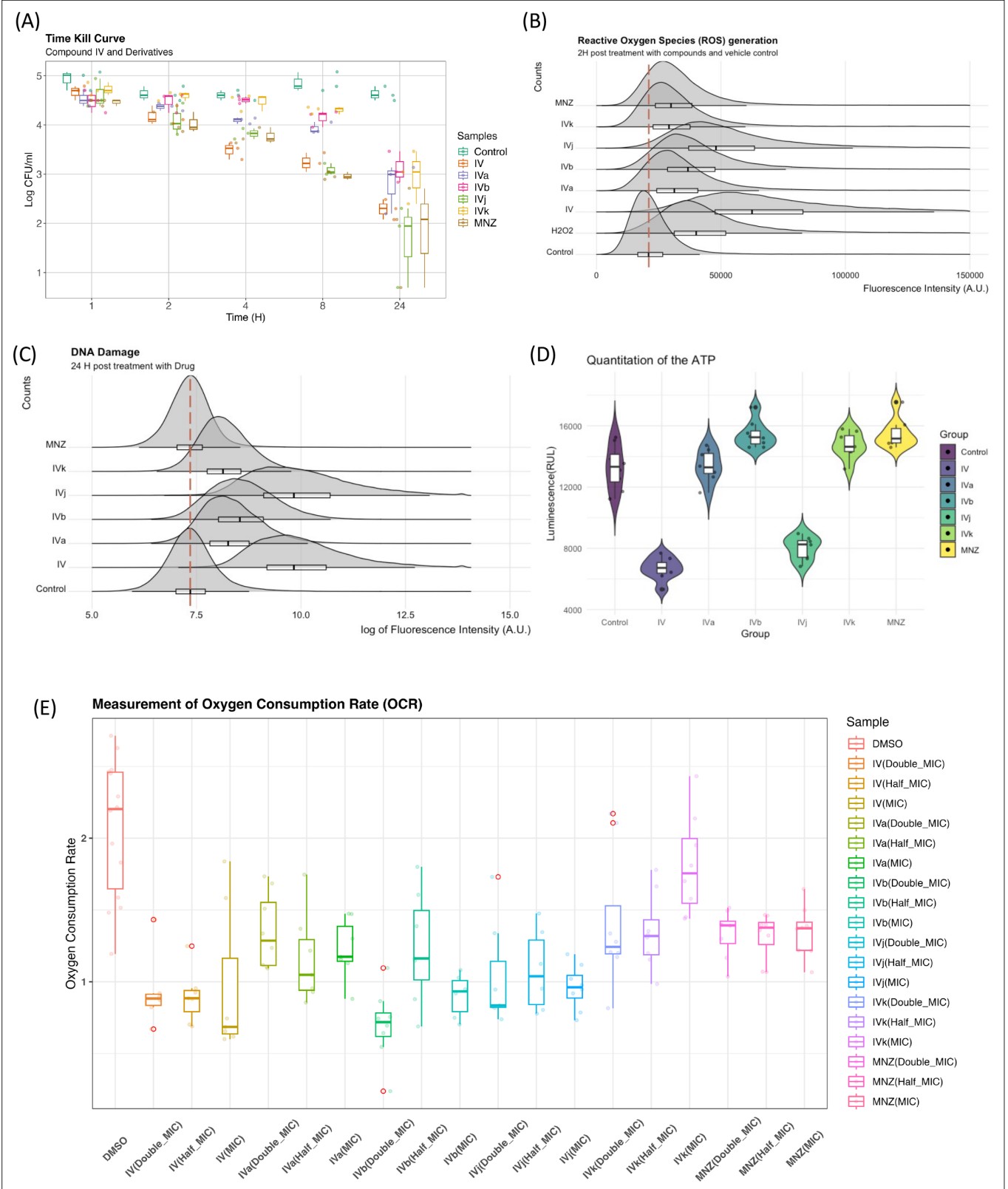

**Figure 6.** Biophysical assays validating changes in the targeted pathway. (**A**) The time-kill curves were obtained by measuring the bacterial growth inhibition using an initial inoculum of 105 CFU/ml at various time points up to 24 hr. We observed bactericidal activity with all of our compounds. (**B**) Flow cytometry analysis was utilized to quantify the generation of reactive oxygen species (ROS), with fluorescence intensity plotted against counts. The presented data represent the compilation of three independent experiments. (**C**) Flow cytometry analysis was performed on both control and

*Figure 6 continued on next page*

*Figure 6 continued*

drug-treated *H. pylori* cells following the TUNEL (terminal deoxynucleotidyl transferase dUTP nick end labeling) assay protocol. The data are presented as a density plot of fluorescence intensity versus the cell counts. The presented data represent the compilation of three independent experiments. (**D**) Changes in the ATP production rate and the presented data represent eight replicates of a single experiment. (**E**) Oxygen consumption rate (OCR) of each sample (with six replicates) at 60 min was related to the basal OCR and compared with the same ratio obtained with the sole DMSO (as the untreated reference sample). Values are the means of the replicates ± standard deviation (SD).

The online version of this article includes the following figure supplement(s) for figure 6:

**Figure supplement 1.** Alteration in solubility of flavodoxin in the Proteome Integral Solubility Alteration (PISA) assay associated with different compounds at two different concentrations.

**Figure supplement 2.** Expression of gene associated with oxidative stress and DNA damage.

## Discussion

Multi-omics have been extensively used in the understanding of host–pathogen interaction and microbiome studies but its full potential for determining antibacterial modes of action has not yet been explored. Modes of action of antibacterial compounds have been typically investigated using targeted experiments focusing on specific cellular processes or pathways, which may not capture the full complexity of antibacterial effects or potential resistance mechanisms. Our results show that multi-omics investigation combined with PISA is a useful tool for the discovery of new previously undetected drug-associated potential targets and for the validation of previously described ones.

We have used here a series of drugs that were initially developed to block Hp-Fld in vitro to showcase a multi-omics-based method for pathway deconvolution that can overcome two main challenges in pathway discovery. First, the detection of new potential targets for feature tailoring of an existing molecule, and second, the validation of the intracellular potential targets previously intended for a specific drug. Traditionally, the antimicrobial drug discovery process focused on identifying essential genes and proteins which could be targeted by inhibitory drugs discovered by phenotype screening. However, there is growing interest in non-essential genes or proteins involved in virulence, quorum sensing, biofilm formation, or immune response evasion. This study identifies four out of our five compounds that induce significant change in the solubility of CagA, the major virulence factor of *H. pylori*. The network analysis of CagA-associated proteins indicates systemic regulation and opens up new possibilities for exploring the efficacy of these drugs in the regulation of the type four secretion system. Nonetheless, all our compounds, except for Compound IVk, additionally target multiple essential proteins. Among the most promising ones is FtsZ, a prominent target in three out of the five compounds tested. Although FtsA has not been categorized as essential, there is direct evidence for the localization of this protein at the cell division site. Along with FtsZ, we have identified FtsA as a potential target for Compounds IV, IVb, and IVj. Our investigation into the systemic changes induced by this interaction spread across the WGCNA module 'brown' and module 'pink'. In the case of Compounds IV and IVj, we have observed a positive correlation with module 'brown', and for Compound IVb, a positive correlation with module 'pink'. Another essential protein target that has been detected across the board is AtpD, a protein belonging to the ATP-synthase complex. In the case of Compound IVj, we have detected four other potential targets from the ATP-synthase complex, namely AtpB, AtpC, AtpF, and AtpG along with AtpD. This poses IVj as a potentially effective ATP-synthase blocker. The ability of IV and IVj to inhibit ATP synthesis has been confirmed and demonstrated by a significant reduction in ATP production (see *Figure 6D*).

Regarding the validation of the pre-defined target flavodoxin, it is paramount to mention that some targets exhibit very limited shifts in solubility changes upon drug binding (*Molina et al., 2013*; *Sabatier et al., 2022*; *Savitski et al., 2014*), making the detection of the target difficult at a statistically significant level. In this regard, we have observed shifts in the solubility of flavodoxin for all the compounds, however, only in the case of Compound IVj the change was statistically significant (*Figure 6—figure supplement 2*). Other limitations of the target discovery process are the poor knowledge of the binding affinity of the compounds to their relative targets and the lack of any structural information about the potential binding site(s). At the methodological level, this research exemplifies the improved resolution of multi-omics data processing through the incorporation of the target deconvolution method. In summary, the study elucidates how the amalgamation of various techniques

can foster a comprehensive grasp of intracellular events, thereby aiding in the advancement of anti-microbial drug development.

## Methods

### Culturing of *H. pylori*

*H. pylori* (ATCC 700392) was purchased from the American Type Culture Collection (ATCC, Manassas, VA, USA). Cultures of *H. pylori* were grown in brain heart infusion (BHI) broth (Oxoid) supplemented with 4% fetal bovine serum (Pan-Biotech, Aidenbach, Germany) under microaerophilic conditions (85% $N_2$, 10% $CO_2$, 5% $O_2$) at 37°C for 72 hr. Prior to any assay, the absence of contaminations was verified using MALDI Biotyper (MBT) microbial identification system.

### Analysis for differentially expressed genes (transcriptomics)

*H. pylori* (ATCC 700392) was grown as above and subjected to further incubation with Compounds IV, IVa, IVb, IVj, IVk, and MNZ at their respective minimum inhibitory concentrations (MICs) for 4 hr. The incubation time for transcriptomics and proteomics assays was determined based on the time-kill curves assay (*Figure 6A*). The 4-hr time point shows a significant amount of cell death compared to the control population. Then, each *H. pylori* culture was washed three times with RNAlater (Sigma-Aldrich) for RNA extraction and library preparation and stored at −80°C. Three independent experiments were performed for each drug (*Figure 1—figure supplement 1B*). The samples were sequenced under Illumina NovaSeq, 2 × 150 bp configure ration. After investigating the quality of the raw data, sequence reads were trimmed to remove possible adapter sequences and nucleotides with poor quality using Trimmomatic v.0.36. The trimmed reads were mapped to the *H. pylori* reference genome available on ENSEMBL using the STAR aligner v.2.5.2b (*Dobin et al., 2013*). Binary alignment and map files were generated as a result of this step. Unique gene hit counts were calculated by using the feature Counts from the Subread package v.1.5.2 (*Liao et al., 2013*). Unique reads that fell within coding regions were counted. After the extraction of gene hit counts, the gene hit counts table was used for downstream differential expression analysis. A table of the top-ranked genes was extracted from a linear model fit using a Limma package (*Ritchie et al., 2015*). Genes with adjusted p-values <0.001 and log fold change of >1 were defined as differentially expressed genes in each comparison.

To identify gene co-expression modules, we performed WGCNA (*Langfelder and Horvath, 2008*) following the normalization of the data using the regularized log transform (rlog) function from the DESeq2 package (*Love et al., 2014*). A scale-free signed network was constructed with a power of 16, which was determined using the Soft Threshold selection method within WGCNA. Genes were then assigned to modules using topological overlap measures and closely related modules were merged based on their Eigengenes. Finally, we investigated correlations between modules and drugs using the Pearson correlation analysis.

### Quantitative proteomics of total protein amount changes

*H. pylori* (ATCC 700392) was grown as above and subjected to further incubation with Compounds IV, IVa, IVb, IVj, IVk, and MNZ at their respected MIC for 4 hr under the same conditions. Next, the *H. pylori* culture was washed three times with phosphate-buffered saline (PBS) to remove any residual protein from the culture media and cell pellets stored at −80°C for protein extraction. The same bacterial mother flask was split into different flasks for producing biological replicates of compound treatment incubation, four for each condition. In order to include all samples, two 16-plex experiments were run with Compounds IV, IVa, and IVb in one plex, and IVj, IVk, and MNZ in the other plex, both together with the same four replicates of DMSO control in each plex, used as identical peptide digested sample before tandem mass tag (TMT) labeling (*Figure 1—figure supplement 1C*). Briefly, samples extracted by incubation for 30 min at 23°C with PBS-based buffers containing 1× protease inhibitor, 50 μg/ml lysozyme, 250 U/ml benzonase, 1 mM $MgCl_2$, and 0.4% NP40. The incubation was followed by adding the same volume of a 2× RIPA buffer, five freeze–thaw cycles using liquid nitrogen and 37°C, and ten 3 s cycles of probe sonication on ice with 3-s stop intervals. The total protein concentration of extracted samples was measured using micro-BCA (bicinchoninic acid) assay and 50 μM of protein was processed by dithiothreitol reduction, iodoacetamide alkylation, and cold acetone precipitation. Samples were resuspended in EPPS (4-(2-Hydroxyethyl)-1-piperazi

nepropanesulfonic acid) buffer, 8 M urea, pH 8.0, diluted down to 4 M urea and digested by LysC, then diluted down to 1 M urea and digested with trypsin. Each digest was labeled using TMTpro 16-plex technology (Thermo Fischer) and a final multiplex sample was first cleaned by Sep-Pack C18 column (Waters). The final sample was fractionated off-line by capillary reversed phase chromatography at pH 10 into 12 fractions, and each of them was then analyzed by high-resolution nLC–ESI-MS/MS (nanoscale liquid chromatography-electrospray ionisation-tandem mass spectrometry) using an Orbitrap Exploris 480 instrument (Thermo Scientific). Peptide and protein identification, and quantification were carried out using Proteome Discoverer 2.5 (Thermo Scientific) with database search against the Uniprot database UP0000000429. Datasets were cleaned and normalized on total TMT ion reporter intensity and on the average of DMSO controls. Then the normalized data were fitted to linear models using Limma (*Ritchie et al., 2015*) and differential expression proteins were selected compared to the DMSO controls.

## Target deconvolution (PISA-Express assay)

*H. pylori* (ATCC 700392) was cultured as described above for 72 hr. The target deconvolution method aims to evaluate the initial interaction with intracellular proteins. We selected a 20-min time point based on intracellular ROS generation (not shown). It is a well-reported phenomenon that bactericidal drugs induce early production of ROS (*Kohanski et al., 2007*). The culture was then incubated with Compounds IV, IVa, IVb, IVj, and IVk at their respective MICs and five times the MIC for 20 min, also under the indicated microaerophilic conditions. For each compound, one PISA-Express experiment was carried out with three biological replicates of a control and each compound concentration. After the incubation, cells were pelleted at 7000 × *g* for 5 min, washed with PBS, and centrifuged. The bacterial pellet was resuspended in 1.6 ml of PBS, and protease inhibitor. For each PISA experiment (*Figure 1—figure supplement 1D*), each replicate was divided into 16 aliquots of 55 µl each, which were processed at 16 different temperatures ranging from 46 to 70°C. The remaining volume in each tube was left for the expression proteomics control of each individual sample for total protein abundance in each individual PISA sample measured in its soluble protein abundance. Each expression proteomics control was treated similar to the corresponding PISA sample, except for thermal treatment and ultracentrifugation. For each PISA sample, after thermal treatment, all temperatures of each replicate were combined and incubated for 30 min at 23°C, with the addition of 50 µg/ml lysozyme, 250 U/ml benzonase, 1 mM MgCl$_2$, and 0.4% NP40. The incubation was followed by five freeze–thaw cycles using liquid nitrogen and 37°C. Ultracentrifugation was performed for 30 min at 125,000 × *g* at 4°C. The protein concentration of supernatants and protein extracts from the fraction of each sample not subjected to thermal treatment was measured using micro-BCA and 50 µg of total protein amount was processed as previously described for 18-plex PISA assay using TMTpro (Thermo Fisher) (*Zhang et al., 2022*). The final sample was fractionated off-line at high pH into 24 fractions, which were then run on nLC–ESI-MS/MS using a high-resolution Orbitrap Exploris 480 instrument.

Peptide and protein identification, and quantification were carried out using Proteome Discoverer 2.5 (Thermo Scientific) with database search against the Uniprot database UP0000000429. Datasets were normalized on total TMT ion reporter intensity and adjusted for batch effect using ComBat (*Zhang et al., 2020*). These normalized data were fitted to linear models using Limma (*Ritchie et al., 2015*) and differential expression proteins were selected compared to the DMSO controls.

## Time-kill curves

*H. pylori* (ATCC 700392) was grown as above at a final cell density of 10$^5$ cells/ml. A sample of 2.5 ml of this pre-inoculum were transferred to each well of a 24-well plate. The culture was subjected to incubation with Compounds IV, IVa, IVb, IVj, and IVk at their respected MIC for 24 hr. At every time point, 20 µl from each condition were sampled from 24-well plate and serially 10-fold diluted in 180 µl of PBS (Millipore). Then, 2.5 µl of each diluted and non-diluted sample were plated on blood agar plates. Colony-forming units were determined by colony counting after 5 days of incubation. Experiments were performed in duplicate and repeated at least twice.

## Cellular ROS assay

The relative amount of intracellular ROS was quantified by using the cell-permeable fluorescent probe 2′,7′-dichlorofluorescin diacetate (DCFDA), which gets de-esterified intracellularly and turns to

highly fluorescent 2',7'-dichlorofluorescein upon oxidation. *H. pylori* was grown for 72 hr with agitation under microaerophilic conditions (85% $N_2$, 10% $CO_2$, 5% $O_2$), harvested, and washed with PBS. Cells were resuspended until a final density of 0.1 in Gibco FluoroBrite DMEM (Dulbecco's Modified Eagle Medium) media supplemented with DCFDA (5 µM). After 30 min of pre-incubation at 37°C, Compounds IV, IVa, IVb, IVj, IVk, MNZ (at their respective MIC) and 20 µM $H_2O_2$ were added, and the fluorescence was measured using a Beckman Coulter flow cytometry system. The DCFDA probe, upon oxidation, emits fluorescent signals that are indicative of intracellular ROS levels. To ensure reliability and generate statistically significant data, the experiment was performed three times independently. The data obtained from each experiment were merged, and ggplot, a data visualization package in the R programming language, was used to visualize the results.

## Compound-induced DNA damage assay

*H. pylori* was grown for 72 hr with agitation under microaerophilic conditions harvested and washed with PBS. Cells were resuspended until a final density of 0.6 in Gibco FluoroBrite DMEM media supplemented with DCFDA (5 µM). After 30 min pre-incubation at 37°C, Compounds IV, IVa, IVb, IVj, IVk, MNZ (at their respective MIC) and 20 µM $H_2O_2$ were added and kept for 24 hr at 37°C under microaerophilic conditions. Next, cells were labeled according to the Apo-direct BD kit protocol. To detect compound-induced DNA breaks, reagent TdT (which attach a bromolated deoxyuridine triphosphate (Br-dUTP) molecule to the damaged DNA that have a free hydroxyl (OH) group) was added to the cells. After treatment with Br-dUTP, the cells were stained with a fluorescent FITC (5/6-fluorescein isothiocyanate)-labeled anti-BrdU monoclonal antibody. This stain helps visualize the damaged DNA in the cells. The cells were then analyzed using flow cytometry.

## Measurement of ATP depletion in *H. pylori*

The ATP depletion in *H.pylori* cells was assessed using a luminescence-based method. Specifically, the BacTiter-Glo microbial cell viability assay kit (Promega, USA) was employed for this purpose. The assay was performed following the manufacturer's instructions. To prepare the BacTiter-Glo reagent, the lyophilized BacTiter-Glo enzyme/substrate mixture was combined with the buffer at room temperature. Separate wells of an opaque-walled 96-well microplate were designated for the drug-treated *H. pylori* culture and the control (DMSO). One hundred microliters of the drug-treated culture and the untreated control were added to their respective wells. Next, 100 µl of the prepared BacTiter-Glo reagent was added to each well containing the *H. pylori* culture and control. The contents were gently mixed and incubated in the dark for 5 min to allow for cell lysis and ATP release. Following the incubation period, luminescence intensity was measured using a Varioskan LUX multimode microplate reader (Thermo Fisher, USA). The luminescence signal is directly proportional to the amount of ATP present in the sample. A decrease in luminescence intensity in the drug-treated samples compared to the untreated control indicates ATP depletion in the *H. pylori* cells after the 2-hr incubation with the drug. The relative ATP depletion can be calculated as the percentage decrease in luminescence compared to the untreated control.

## Measurement of OCR

Cellular OCR was measured using the extracellular flux analyzer XFe96 (from Seahorse Bioscience at the MOSBRI EU Infrastructure – HypACB facility at Sapienza University). To reproduce the biological condition in the stomach, where oxygen levels are relatively low this assay was conducted under hypoxic conditions (*Abass et al., 2021*). *H. pylori* cells were cultured in BHI at 37°C (HypoxicLab, Oxford Optronic-95% $N_2$, 5% $O_2$, 10% $CO_2$, 20% humidity), then transferred into the Don Whitley (i2 workstation) hypoxic chamber (95% $N_2$, 5% $O_2$, 20% humidity), where cellular OCR was measured.

The sensor cartridge for Seahorse was hydrated at 37°C a day before each experiment, following the manufacturer's instructions, under hypoxic conditions. Before cell seeding, the Seahorse 96-well plates were coated by adding 20 µl of a 1× collagen solution in each well, which was prepared from a 20× stock type I solution (Sigma C3867-1VL). After 1 hr, the collagen solution was removed, and the wells were left to dry under a laminar flow hood. The last column of the plate was not coated since it contained the medium alone to allow the addition of sodium sulfite (VWR 0628-500g) during the run. Sodium sulfite (100 mM final concentration) was used as a chemical oxygen scavenger to provide a 'zero' oxygen reference. Coating plus medium was used to measure the environmental oxygen levels

as a reference during the experiment. The injection ports containing the different stressors/additives (i.e., 10× DMSO, 10× stock drugs, or 10× sulfite) were loaded (20 μl per well) in the Seahorse chamber, 1 hr before the Seahorse run.

Exponentially growing *H. pylori* cells were harvested by centrifugation (10 min at 4000 rpm) and resuspended in the growth chamber with Seahorse medium (supplemented with 10 mM glucose, 1 mM pyruvate, and 2 mM glutamine, following the manufacturer's instructions) to achieve a final $OD_{600}$ of 0.6. The cells were then seeded on the collagen-coated plate (180 μl per well), except for the last column, and vortexed. After a Seahorse calibration run, the cell plate was loaded into the instrument following the manufacturer's instructions.

Measurements were set as follows: 3 min of mixing, where the plate well was allowed to exchange gases with the hypoxic environment, followed by 3 min of measurements. During the 3-min measurement, the 96 probes protruding from the lid were automatically inserted into the cell plate, creating a microchamber per well. This transiently sealed microchamber allowed the instrument to measure the oxygen consumption carried out by the cell monolayer without any gas exchange with the environment. After six baseline measurements, the drug/sulfite was automatically added to each well, as indicated. Kinetics were considered for the first hour to avoid any bias due to cell stress or excessive metabolic activity.

To compare the different OCR obtained from independent experiments, relative values have been used as follows: OCR of each sample (with six replicates) at 60 min was related to the basal OCR and compared with the same ratio obtained with the sole DMSO (as the untreated reference sample). Values are the means of the replicates ± standard deviation.

## Acknowledgements

Full support in the experimental design, performance, and data analysis of the proteomic study was provided by the 'Chemical Proteomics' Unit of Karolinska Institutet (KI) within the Division of Chemistry I at the Department of Medical Biochemistry and Biophysics. Chemical Proteomics is a KI core facility, a national unit of SciLifeLab and a national node of the Swedish National Infrastructure for Biological Mass Spectrometry (BioMS). We would also like to express our gratitude to Dr. Ana Isabel López and her team at Hospital Universitario Miguel Servet, Zaragoza, as well as Prof. Volkan Özenci and Ramona Santini at Karolinska Institutet, for their continuous assistance in verifying microbial samples using the MALDI Biotyper (MBT) microbial identification system. We also thank the Molecular Scale Biophysics Research Infrastructure (MOSBRI) for transnational access funding. This project has received funding from the European Union's H2020 research and innovation programme under Maria Sklodowska-Curie grant agreement No 801586. We acknowledge financial support to JS from FLAV4AMR (JPIAMR), European Union's Horizon 2020 (MOSBRI, grant 101004806), PID2019-107293GB-I00 and PID2022-141068NB-I00 (MICINN, Spain), and E45_23R (Gobierno de Aragón, Spain).

## Additional information

### Funding

| Funder | Grant reference number | Author |
|---|---|---|
| H2020 Marie Skłodowska-Curie Actions | 10.3030/801586 | Ritwik Maity<br>Javier Sancho |
| Joint Programming Initiative on Antimicrobial Resistance | FLAV4AMR | Javier Sancho |
| Horizon 2020 Framework Programme | MOSBRI grant 101004806 | Javier Sancho |
| Horizon 2020 Framework Programme | PID2019-107293GB-I00 | Javier Sancho |
| Ministerio de Ciencia, Innovación y Universidades | PID2022-141068NB-I00 | Javier Sancho |

| Funder | Grant reference number | Author |
|---|---|---|
| Gobierno de Aragón | E45_23R | Javier Sancho |

The funders had no role in study design, data collection, and interpretation, or the decision to submit the work for publication.

## Author contributions

Ritwik Maity, Conceptualization, Resources, Data curation, Software, Formal analysis, Validation, Investigation, Visualization, Methodology, Writing – original draft, Writing – review and editing; Xuepei Zhang, Francesca Romana Liberati, Data curation, Formal analysis; Chiara Scribani Rossi, Francesca Cutruzzolá, Data curation, Methodology; Serena Rinaldo, Massimiliano Gaetani, Data curation, Formal analysis, Methodology; José Antonio Aínsa, Supervision, Methodology, Writing – original draft, Project administration; Javier Sancho, Conceptualization, Resources, Software, Formal analysis, Supervision, Funding acquisition, Validation, Investigation, Visualization, Methodology, Writing – original draft, Project administration, Writing – review and editing

## Author ORCIDs

Ritwik Maity 
Chiara Scribani Rossi 
Serena Rinaldo 
José Antonio Aínsa 
Javier Sancho 

Reviewer #1 (Public Review): https://doi.org/10.7554/eLife.96343.3.sa1
Author response https://doi.org/10.7554/eLife.96343.3.sa2

---

# Additional files

## Supplementary files

• MDAR checklist

## Data availability

Dataset for publication: Merging Multi-OMICs with Proteome Integral Solubility Alteration Unveils Antibiotic Mode of Action [Dataset]. Zenodo. https://doi.org/10.5281/zenodo.8321088. This data set contains Count files from RNA-seq, mass spectrometry proteomics, Proteome Integral Solubility Alteration and data from Cytoscape. The record is publicly accessible, but files are restricted to users with access.

The following dataset was generated:

| Author(s) | Year | Dataset title | Dataset URL | Database and Identifier |
|---|---|---|---|---|
| Maity R, Zhang X, Liberati FR, Rossi CS, Cutruzzolà F, Gaetani M, Aínsa JA, Sancho J | 2023 | Dataset for publication: Merging Multi-OMICs with Proteome Integral Solubility Alteration Unveils Antibiotic Mode of Action | https://zenodo.org/records/8321088 | Zenodo, 10.5281/zenodo.8321087 |

---

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
