## [Editor Report · eLife assessment]

This **fundamental** study provides insights into how pathogens respond, on a systemic level including several gene targets and clusters, to selected antimicrobial molecules. **Compelling** evidence is provided, through multi-omics and functional approaches, that very similar molecules originally designed to target the same bacterial protein act differently within the context of the whole set of cellular transcripts, expressed proteins, and pre-lethal metabolic changes. Given the rapid accumulation of omics data and the much slower capacity of extracting biologically relevant insights from big data, this work exemplifies how the development of sensitive data analysis is still a major necessity in modern research.

---

## [Referee Report · Reviewer #1 (Public Review)]

In this manuscript, entitled " Merging Multi-OMICs with Proteome Integral Solubility Alteration Unveils Antibiotic Mode of Action", Dr. Maity and colleagues aim to elucidate the mechanisms of action of antibiotics through combined approaches of omics and the PISA tool to discover new targets of five drugs developed against Helicobacter pylori.

Strengths:

Using transcriptomics, proteomic analysis, protein stability (PISA), and integrative analysis, Dr. Maity and colleagues have identified pathways targeted by five compounds initially discovered as inhibitors against H. pylori flavodoxin. This study underscores the necessity of a global approach to comprehensively understand the mechanisms of drug action. The experiments conducted in this paper are well designed and the obtained results support the authors' conclusions.

---

## [Author Response]

The following is the authors’ response to the original reviews.

**Public Reviews:**

**Reviewer #1 (Public Review):**
In this manuscript, entitled " Merging Mul-OMICs with Proteome Integral Solubility Alteration Unveils Antibiotic Mode of Acon", Dr. Maity and colleagues aim to elucidate the mechanisms of action of antibiotics through combined approaches of omics and the PISA tool to discover new targets of five drugs developed against Helicobacter pylori.Strengths:Using transcriptomics, proteomic analysis, protein stability (PISA), and integrative analysis, Dr. Maity and colleagues have identified pathways targeted by five compounds initially discovered as inhibitors against H. pylori flavodoxin. This study underscores the necessity of a global approach to comprehensively understanding the mechanisms of drug action. The experiments conducted in this paper are well-designed and the obtained results support the authors' conclusions.Weaknesses:This manuscript describes several interesting findings. A few points listed below require further clarification:(1) Compounds IVk exhibits markedly different behavior compared to the other compounds. The authors are encouraged to discuss these findings in the context of existing literature or chemical principles.

This is a good point. We have added the following paragraph (Page No-13).

*“In several of our studies, compound IVk, which has a higher MIC, exhibits markedly different behavior. This difference in behavior may stem from different sources, including intercellular availability, inactivation inside the cell, or loss of target specificity. Multiple studies have previously demonstrated that there is only a 30% chance for a structurally similar compound to have similar biological activity32.”*
(2) The incubation me for treating H. pylori with the drugs was set at 4 hours for transcriptomic and proteomic analyses, compared to 20 min for PISA analysis. The authors need to explain the reason for these differences in treatment duration.

This is now explained in Pages 17 and 19, where the following paragraphs have been included

*“The target deconvolution method aims to evaluate the initial interaction with intracellular proteins. We selected a 20-minute time point based on intracellular ROS generation (not shown). It is a well-reported phenomenon that bactericidal drugs induce early production of ROS.”*

*“The incubation time for transcriptomics and proteomics assays was determined based on the Time-Kill Curves assay (Fig. 6(A)). The 4-hour time point shows a significant amount of cell death compared to the control population.”*
(3) The PISA method facilitates the identification of proteins stabilized by drug treatment. DnaJ and Trigger factor (g), well-known molecular chaperones, prevent protein aggregation under stress. Their enrichment in the soluble fraction is expected and does not necessarily indicate direct stabilization by the drugs. The possibility that their stabilization results from binding to other proteins destabilized by the drugs should be considered. To prevent any misunderstanding, the authors should clarify that their methodology does not solely identify direct targets. Instead, the combination of their findings sheds light on various pathways affected by the treatment.

This is also a very valuable observation. We now clearly state that in new paragraphs at Pages 8 and 13

*Another target shared among several compounds is the chaperone protein trigger factor (Tig), which plays a crucial role in facilitating proper protein folding and is indispensable for the survival of bacterial cells. The solubility of this protein has been altered by all the compounds except IVk (Fig. 2(I-J)) in a concentration-dependent manner (Fig. S4(B, D, and E)). The possibility of Tig interacting with other proteins destabilized by the drug, along with the influence of the heat gradient during the PISA assay, may introduce potential noise in the data. Further investigation is required to confirm the interaction of the drug with Tig.*

*“The module “black” associated with this compound contains Tig, which is involved in facilitating proper protein folding, as a target, and it down-regulates multiple proteins associated closely with S12 ribosomal protein of the 30S subunit (Fig. S9(D)) indicating its involvement in stabilization of ribosomal protein.”*

(4) At the end of the manuscript, the authors conclude that four compounds "strongly interact with CagA". However, detailed molecule/protein interaction studies are necessary to definitively support this claim. The authors should exercise caution in their statement. As the authors mentioned, additional research (not mandated in the scope of this current paper) is necessary to determine the drug's binding affinity to the proposed targets.

We have modified the sentence (Page -15) to say:

*“This study identifies four out of our five compounds that induce significant change in the solubility of CagA, the major virulence factor of H. pylori.”*

(5) The authors should clarify the PISA-Express approach over standard PISA. A detailed explanation of the differences between both methods in the main text is important.

This was already explained in Page 5 (no changes have been made)

**Reviewer #2 (Public Review):**
Summary:This work has an important and ambitious goal: understanding the effects of drugs, in this case antimicrobial molecules, from a holistic perspective. This means that the effect of drugs on a group of genes and whole metabolic pathways is unveiled, rather than its immediate effect on a protein target only. To achieve this goal the authors successfully implement the PISA-Express method (Protein Integral Solubility Alteration), using combined transcriptomics, proteomics, and drug-induced changes in protein stability to retrieve a large number of genes and proteins affected by the used compounds. The compounds used in the study (compound IVa, IVb, IVj, and IVk) were all derived from the precursors compound IV, they are effective against Helicobacter pylori, and their mode of action on clusters of genes and proteins has been compared to the one of the known pylori drug metronidazole (MNZ). Due to this comparison, and confirmed by the diversity of responses induced by these very similar compounds, it can be understood that the approach used is reliable and very informative. Notably, although all compound IV derivatives were designed to target pylori Flavodoxin (Fld), only one showed a statically significant shift of Fld solubility (compound IVj, FIG S11). For most other compounds, instead, the involvement of other possible targets affecting diverse metabolic pathways was also observed, notably concerning a series of genes with other important functions: CagA (virulence factor), FtsY/FtsA (cell division), AtpD (ATP-synthase complex), the essential GTPase ObgE, Tig (protein export), as well as other proteins involved in ribosomal synthesis, chemotaxis/motility and DNA replication/repairs. Finally, for all tested molecules, in vivo functional data have been collected that parallel the omics predictions, comforting them and showing that compound IV derivatives differently affect cellular generation of reactive oxygen species (ROS), oxygen consumption rates (OCR), DNA damage, and ATP synthesis.Strengths:The approach used is very potent in retrieving the effects of chemically active molecules (in this case antimicrobial ones) on whole cells, evidencing protein and gene networks that are involved in cell sensitivity to the studied molecules. The choice of these compounds against H. pylori is perfect, showcasing how different the real biological response is, compared to the hypothetical one. In fact, although all molecules were retrieved based on their activity on Fld, the authors unambiguously show that large unexpected gene clusters may, and in fact are, affected by these compounds, and each of them in different manners.Impact:The present work is the first report relying on PISA-Express performed on living bacterial cells. Because of its findings, this work will certainly have a high impact on the way we design research to develop effective drugs, allowing us to understand the fine effects of a drug on gene clusters, drive molecule design towards specific metabolic pathways, and eventually better plan the combination of multiple active molecules for drug formulation. Beyond this, however, we expect this article to impact other related and unrelated fields of research as well. The same holistic approaches might also allow gaining deep, and sometimes unexpected, insight into the cellular targets involved in drug side effects, drug resistance, toxicity, and cellular adaptation, in fields beyond the medicinal one, such as cellular biology and environmental studies on pollutants.
**Recommendations for the authors:**

**Reviewer #1 (Recommendations For The Authors):**
Please modify these few concerns:- It is unclear from the introduction and discussion whether conventional transcriptomic and proteomic analyses have previously been conducted on the compounds examined in this study. If only targeted studies have been performed please clarify this further.

To make it more clear, we have added the following paragraph in Page 5:

*“Our investigation into understanding the mode of action of nitro-benzoxadiazole compounds commenced with a comparison of the conventional transcriptional and translational changes induced by these compounds, the vehicle control (DMSO), and the commercially used drug MNZ. RNA sequencing (RNA-seq) and expressional proteomics were employed to identify transcriptional and translational changes, respectively.”*

- The decision to monitor the oxygen consumption rate (OCR) is based on the hypothesis that the drugs would impact flavodoxins function. Could the authors cite specific studies that suggest a reduction in flavodoxin leads to decreased OCR that can be measured?

The reviewer is correct to say that we have done this study based on our hypothesis that a reduction in flavodoxin may lead to decreased OCR. To our knowledge, there is no previous studies indicating that so we now clearly state (Page 14) that it is our hypothesis.

*“On the other hand, given that these drugs indicated involvement of multiple factors from the electron transport chain including flavodoxin and we observed significant drop in the ATP production rate (Fig. 6(D)) associated to compounds IV and IVj, we have investigated the changes in oxygen consumption rate (OCR) as we hypothesize that a reduction in soluble flavodoxin could lead to decreased OCR”*

- Increase font size in some figures and supplemental materials for clarity.

We acknowledge the reviewer's comment and have addressed it to the best possible extent in the figures.

- Correct figure references throughout the text (example of mistake p4, Fig S1D, p6 S1C).

We have corrected the figure references.

- Check spelling errors, for example, Figure S1B: "library preparation".

We have revised the figures and corrected spelling errors.

- Ensure H. pylori is in italics.

Done!

- Figure S4: Replace (D) by (E).

Done! Thank you.

- Page 7: Check the sentence: "...RpleE, InfC and F Furthermore, we..." .

Corrected!

*“The 20 common essential targets are mostly associated with cell division (for example, FtsZ), small subunit ribosomal proteins (RspC, RspE, RspL, RplE, InfC). Furthermore, we identified a few unique changes for compound IV (DnaN, involved in DNA tethering and processivity of DNA polymerases, and C694_06445, which could be a functional equivalent of delta subunit of DNA polymerase III).”*

- Page 9: Please modify the name of one compound "Compounds IV, IVj (and not IVk) and MnZ downregulate...".

We have observed that both reviewers mentioned this point and we revisited the data, as suggested by Fig S8(B), that compounds IV, IVk, and MNZ cluster together and downregulate the genes associated with this pathway. Based on this, we have not changed anything in the text.

- Figure S9: please clarify symbols (triangles and others) in the Figure legend.

Done!

- Page 9: Is it the Figure S9B you are referring to? Talking about proteomics?

Sorry, we have not understood the above comment.

**Reviewer #2 (Recommendations For The Authors):**
All figures are printed as one per page. In this format, almost all pictures suffer a severe problem with dimensions. Notably graph axes and axis values, subtitles, and legends within the pictures are too small, although the graphical part is almost always appropriate. Negative example (higher fonts are needed): Figure 1. Positive example (font ok): Figure 2A or Figure 3 right panels.

We have carefully revised our figures to address the issues you mentioned, ensuring that elements are visible when printed one per page. In Fig 1: We have increased the font sizes of the graph axes, axis values, subtitles, and legends to improve readability. Additionally, we have color-matched different Gene Ontology (GO) terms for better rideability. In Fig 2: To enhance clarity, we have resized the figure by removing the top 10 protein list, now presented in a separate table. This ensures that the figure's main content remains prominent. These modifications have been made across figures to maintain consistency and readability.

For all figures, particularly for non-experts, not only a list of what is found in the picture should be provided, but also a minimal, simplified key of interpretation (of what is to be noticed). Particularly relevant for scatter plots.

We have modified the legends to provide simplified key interpretation for the scatter plots.

In general for most analyses I see the involvement of FtsA, whereas most discussions concern FtsY and FtsZ. Maybe this point should be clarified. For example: (i) FtsZ is quoted in the Second "Results" paragraph (page 6), but we can't find this gene in Figure 2, nor in the corresponding table (Figure 2A); (ii) FtsY downregulation is quoted in the Fifth "Results" paragraph (page 9), but we can't find this gene in Figure 5, 9S or 10S.

We are not entirely sure if we have understood the reviewer's comment correctly, as we did not mention FtsY in our discussion section. In the discussion section, we have focused on the involvement of FtsZ and FtsA with some of our compounds. We decided to discuss them together because FtsZ is the primary component that is recruited to the membrane by the actin-related protein FtsA, while the role of FtsY remains highly debated.

Figure 1: same colour for the same GO: term in different panels should be used.

Done!

Figure 4: please specify (being it essential throughout the whole paper) that the group colouring only refers to Figure 4A, lower bar.

Done!

Figure 5, S9, and S10: having the combination of analysed sets (brown / IV , magenta / IVb, etc....) as a panel subtle is almost a necessity, to avoid constant page turning. I did rewrite all of them by hand to be able to follow the main text story.

Done!

What are the triangles? (this is not written anywhere).

We have now explained this in the legends of Fig5.

Figures S9 and S10 are too crowded (please refer to Figure 5 for a good format/size).

For supplementary figures S9 and S10 we prefer to keep the gene names, but in order to make them more legible we have now added subtitles to each panel.

Second and third "Results" paragraph. Explicitly saying that the Second is only focused on TOP 10 hits, at the beginning of the paragraph (while the third on essential genes) would help enormously the non-specialist in orienting among the different sections.

On page 7, we have revised the text to indicate that the paragraph is only focused on the top 10 hits. Additionally, we have included a table of top 10 hits for better clarity and accessibility.

Page 6: the following sentence should be in the introduction, to stress the novelty of the work: "This is the first me PISA assay, in the form of PISA-Express, has been successfully performed in living bacterial cells, with protocols adapted and modified from previous PISA studies in mammalian cells".Page- 2

We agree this is an important point. However, having we stated it in both the abstract and in the PISA section in the results we prefer not to state it once more in the Introduction.

(no changes made)

I couldn't find any reference to Figure S3 in the text.

Included! (P 9)

"Compounds IV, IVk, and MNZ downregulate the genes associated with this pathway (Fig. 4(B) & S8(B))": it seems to me that it is IVj rather than IVk to downregulate. Please check carefully.

We have observed that both reviewers mentioned this point and we revisited the data, as suggested by Fig S8(B), that compounds IV, IVk, and MNZ cluster together and downregulate the genes associated with this pathway. Based on this, we have not changed anything in the text.

Page 12: of the pre-defined target like flavodoxin => of the pre-defined target flavodoxin.

Thanks! We have removed “like” from the sentence.

Metronidazol (=MNZ) only appears on page 13 (MNZ already on page 8).

Corrected! The correspondence is now first indicated in P. 3.

Please resolve the ambiguity metronidazol/metronidazole (main text and figures).

We now always say “metronidazole”

The Sixth "Results" paragraph (pages 10-11) should be developed a bit more. All Figure 6 results are summarized in 8 lines at the end of the paragraph. This doesn't bring much, particularly to a non-specialist reader. Please, for each panel, clearly explain what is to be noticed and what main conclusion(s) can be extracted.

We have improved the description of the section. The modified part now reads:

*…This indicates that the nitro-bearing groups have a higher propensity to generate ROS. We have also observed that the genes associated with the generation of ROS are significantly overexpressed for compounds IV, IVb, IVj, and MNZ (Fig. S12(A)). As described above and depicted in Fig. S12(B), multiple DNA damage repair proteins and genes are down-regulated in the presence of compounds IV, IVb, IVj, and MNZ. Additionally, DNA PolA was found to be a major target for compound IVj. Following these results, we investigated compound-induced DNA damage using the APO BrdU TUNEL assay. All the compounds, particularly IV and IVj, caused significant DNA damage (Fig. 6(C)).*

*On the other hand, given that these drugs indicated involvement of multiple factors from the electron transport chain including flavodoxin and we observed significant drop in the ATP production rate (Fig. 6(D)) associated to compounds IV and IVj, we have investigated the changes in oxygen consumption rate (OCR) as we hypothesize that a reduction in soluble flavodoxin could lead to decreased OCR. Though the signal-to-noise ratio of these data is poor…*

and we added figure S12 for clarity.

In the same section I found: "Compound IV and its derivatives cause a marked increase in ROS generation when compared to the control (DMSO)" => refers to THIS work or previous work? (in the later case, please quote it).

This data is from our current paper, as shown in Fig 6(B).

In the same paragraph, "the signal-to-noise ratio of these data is considerable" => does it mean that you have good (high signal-to-noise) data, or that you have too high noise for precise quantification? I rather understood the later, but this sentence definitely needs to be rewritten.

Thank you for pointing out the mistake. Your interpretation is correct. We have corrected the sentence.